# Comparative Analysis of Performance Factors between Ladies Professional Golf Association and Korea Ladies Professional Golf Association Golfers

**DOI:** 10.3390/sports10050072

**Published:** 2022-05-03

**Authors:** SunHee Chung

**Affiliations:** Department of Sport Industry Studies, Yonsei University Sports Science Museum, Yonsei University, 50, Seodaemun-gu, Seoul 03722, Korea; gg66088@naver.com

**Keywords:** Ladies Professional Golf Association, Korea Ladies Professional Golf Association, performance factors

## Abstract

This study aimed to analyze the performance factors of both Ladies Professional Golf Association (LPGA) and Korea Ladies Professional Golf Association (KLPGA) players and suggest which performance factors they should improve to play in world-level games. Data from 180 LPGA and KLPGA players who ranked within the top 60 in prize money rankings from 2018 to 2020 were analyzed. Then, the data from LPGA and KLPGA golfers were compared using the seemingly unrelated estimation method. As a result of analyzing 178 data, excluding two players who had missing values, this study found that among LPGA player data, putting average (PA), sand save (SS), green in regulation (GIR), and birdies (BIR) had a significant impact in 2018. Additionally, scoring average (SA) and top-10 finish (T10) had a significant impact in 2019. However, there were no factors influencing performance in 2020.From the results of analyzing 180 players who ranked within the top 60 in KLPGA prize money rankings, there were no performance factors that significantly affected their performance in 2018. However, driver distance (DD) in 2019 and DD and T10 in 2020 affected performance. In conclusion, short games were the most important factor on the LPGA Tour, and driving distance was the most important trend on the KLPGA Tour. Therefore, KLPGA golfers should train in abilities such as putting and ironshots.

## 1. Introduction

The coronavirus (COVID-19) pandemic has driven the whole world into a Public Health Emergency of International Concern (PHEIC) and has been declared to be at the pandemic alert level of Phase 6 [1]. COVID-19 has changed various aspects of life, including politics, economy, society, and culture. The continuous aftereffects of COVID-19 have led to reduced social interaction and strengthened social distancing measures between groups to minimize mass infections. The “new normal” era, symbolic of these mass infections, has become significantly different from prepandemic society [2].There have also been numerous unexpected changes in the field of sports. First, the Tokyo Olympics—initially planned to be held in 2020—were postponed for a year as the COVID-19 pandemic became prolonged; in the 124-year history of the modern Olympic Games, this is the first time that the Summer Olympics were postponed for one year due to a global pandemic crisis [3]. Audiences disappeared from professional sport matches, national-level competitions were cancelled, and sport facilities were either temporarily closed or went out of business.

Amid such adverse conditions, the golf industry continued to look for solutions, while golfers remained active in their roles. For the first time in 112 years—since the second Olympics in St. Louis, 1904—golf was reinstated as an official event in the 2016 Summer Olympics in Rio [4]. Golfers representing their nations participated in the Olympics, where they displayed their exceptional abilities. Gold medals were won by Inbee Park (Korea) in the Rio 2016 Olympics and by Nelly Korda (United States) in the Tokyo 2020 Olympics [5]. Furthermore, professional female golfers from various nations have competed onthe world stage of the Ladies Professional Golf Association (LPGA), winning it and making a name for themselves. Earlier in the history of the LPGA, tour tournaments were led by American athletes; in recent years, numerous international players have demonstrated their skills and lead the LPGA [6]. Since Se-riPak’s victory in the 1998 McDonald’s LPGA Championship, many Korean professional female golfers have actively performed on the LPGA stage; for instance, Jin Young Ko won the Founders’ Cup in October 2021 and tied the record of 14 consecutive rounds in the 60s. In addition, as other Asian and European golfers began to participate in the LPGA, it grew to become a world-class golf tournament.

Golfers’ activities can be statistically verified through performance data, which is related to score average and recorded by Tour officials [7]. Among the Korea Ladies Professional Golf Association’s (KLPGA) performance data, athletic performance factors are represented by variables such as driver distance, fairways in regulation, greens in regulation, putting average, par saves, and recovery; score-related variables are sorted into scoring average, birdies, and par breaks. Other comprehensive factors include rank, top-10 finishes, and number of victories [8]. Meanwhile, LPGA performance factors are classified into six categories (prize money, driving, short game, scoring, total played, and ranking) with subdivisions of 29 subcategories [5]. The “Driving” category is divided into driver distance and driver accuracy, while the “Short Game” category is divided into subcategories such as greens in regulation, putting average, and sand saves. The “Scoring” category is divided into the subcategories of scoring average, top-10 finishes, victories, birdies, eagles, and rounds in the 60s [9,10].

Such data are used to closely analyze teams’ and individual team members’ athletic performances [11,12,13]. In the past, analysis of sports matches mostly consisted of subjective interpretation by the related sport’s field expert.However, Fernandez, Mendez-Villanueva, and Plum mentioned that data analysis regarding the performance of players should be based on scientifically collected evidence and that the analysis of a player’s performance should be based on objective statistic data [14]. Additionally, Hughes and Franks claimed that performance analysis based on objective statistical data could provide meaningful data for coaches to improveplayers’ performances in the future [15]. Specifically, sporting events with active professional scenes, such as golf, basketball, and baseball, maintain objective measures of performance outcome data at both the individual and team levels [16] along with continued research efforts to improve athletes’ performance through data analysis. Among these sports, golf has become popular as its value as a sport has been recognized: Presently, it is not only the professional golfers, but also the amateur athletes and spectators who are interested in professional golfers’ performance [14]. However, as evidenced by the discrepancies between the performance data provided by the KLPGA and the LPGA, the impact of such factors on golfers’ performancemust be verified. In the Information Age, in which the amount of usable data continues to increase, the prediction and analysis of performance through the quantitative analysis of data is of extremely important significance [17].

As of October 2021, the prize money rankings among active athletes in the LPGA indicate that there are 18 American LPGA golfers (36%) and 10 KLPGA golfers (20%) among the top 50; in the top 5, there is oneAmerican LPGA golfer and twoKLPGA golfers. Thus, the athletic performance of golfers who participate in LPGA Tours can be divided largely into two groups: American LPGA and KLPGA golfers. While KLPGA golfers have achieved successful results in the LPGA, it is difficult to find research on each group’s athletic performance factors (LPGA and KLPGA golfers). Existing studies on LPGA and KLPGA golfers include one that suggests early recognition of talent, parents’ commitment to talent training, and sports sponsorships from enterprises as the success factors for KLPGA golfers [18]; one on the racial discrimination experienced by KLPGA golfers [19]; and one on the influence exerted by LPGA events on domestic viewers [6]. Studies on LPGA golfers, including those related to performance, mainly comprise gender, culture, and success stories. Studies on professional golfers’ athletic performance focus on the golfers that belong to the Professional Golfers’ Association (PGA): research on the records of professional golfers’ performance in the PGA, LPGA, and Senior Professional Golf Association (SPGA) by [20]; descriptive research on PGA golfers’ performance between 1992 and 2001 by [21]; and a study on PGA golfers’ competitive skills over the 1998–2001 period by [22].

There has been a lack of research on the performance of LPGA golfers.The previous study about performance factors focused on analyzing factors to improve the scores for both KLPGA and LPGA players or comparing performances for players from both organizations. As KLPGA players were introduced to the LPGA, it was necessary to compare KLPGA andLPGA players’ performances and identify factors that limit success in the LPGA.

Such research material can provide opportunities for golfers to identify not only their own strengths and weaknesses, but also those of their competitors; by establishing the directionality of training in this study, it is expected that golfers’ performance can be strategically improved. Performance analysis based on athletes’ objective statistical data can aid in improving their future performance by providing more meaningful data to the trainers and managers [23]. Furthermore, the current study is necessary in that the comparative analysis of athlete performance outcomes between LPGA and KLPGA golfers can provide objective data not only for amateur golfers watching professional events, but also for professional golfers, coaches, and managers whose goal is to compete in the LPGA and KLPGA, as well as for professional golfers’ sponsors. Specifically, since the environment has changed due to COVID-19, and conventional, familiar, or singular playstyles can no longer be used [24], verifying the effect of this novel environment on golfers’ performance will present training directionalities to golfers of the new normal era.

The purpose of the current study is to identify performance factors that are directly related to professional golfers’ skills, analyze those with an influence on prize money, and predict those that apply to individual golfers. The process will produce the required data for a comparative analysis of performance factors between LPGA and KLPGA golfers. Such an analysis can be used as a foundation for golfers who plan to compete on the LPGA Tour stage to improve their competition outcomes. Competition between golfers from various nations can benefit not only the caliber of individual golfers, but also play a positive role in vitalizing the golf industry as a whole. Specifically, as the COVID-19 pandemic seems to be entering a more stable phase, it is likely that KLPGA golfers will try to advance into the LPGA, the international stage. The current research is necessary since it can provide suggestions for these golfers regarding the areas—or directionality—on which they should be focusing their efforts.

## 2. Methods

### 2.1. Participants

The current study analyzes the performance data of the top 60 golfers for 2018–2020 based on prize money, as recorded and published on the LPGA(www.lpga.com, accessed on 5 November 2021) and KLPGA(www.klpga.co.kr, accessed on 9 December 2021) websites. This is because 95% of competition winners come from the list of the top 60 golfers based on prize money [25], and the fact that they earn seeds for the next year’s competitions. Excluding golfers with missing values in their annual data, data on a total of 178 LPGA golfers were analyzed: 59 golfers for 2018, 59 golfers for 2019, and 60 golfers for 2020. There were no missing values for golfers in the KLPGA, and data on a total of 180 KLPGA golfers were analyzed.

### 2.2. Analysis

To identify factors in the prize money ranking of LPGA golfers, the following performance factors were selected: scoring average (SA), driver distance (DD), driver accuracy (DA), greens in regulation (GIR), putting average (PA), sand saves (SS), birdies (BIR), eagles (EAG), 60-strokes average (60SA), and top-10 finish (T10) (Table 1). Similarly, SA, DD, DA, GIR, PA, BIR, recovery (RE), par saves (PS), par breaks (PB), and T10 were selected as factors in the prize money ranking of KLPGA golfers (Table 2). These factors were mainly used to measure the ability of professional golfers [12,13,26].The computational methods for deriving each of the variables selected for the LPGA and KLPGA are as follows.

Data on individual golfers, provided by the LPGA and KLPGA, were organized using Excel before they were analyzed with the statistical analysis software STATA. To analyze the performance skill factors of the top 60 golfers in the LPGA and KLPGA, the differences in the coefficients were verified using the seemingly unrelated estimation(SUE) method independently from an ordinary least squares (OLS) analysis for estimating the marginal effect of each variable. The popularity of SUE is related to its applicability to a large class of modeling and testing problems and also the relative ease of estimation [27]. The OLS is useful when the parameters are unknown and the relationship between the dependent variable and the explanatory variable is a hypothesis that must be tested [28].This was done to analyze two sets of data from two nations at identical points in time using separate models.The equation of basic SUE is as follows:(1)$$y_{i}=X_{i}\beta_{i}+u_{i},\;i=1,\;\ldots ,\;N$$
(2)$$\left[\begin{array}{c} y_{1}\\ y_{2}\\ \vdots \\ y_{N} \end{array}\right]=\left[\begin{array}{ccc} \begin{array}{cc} X_{1} & 0\\ 0 & X_{2} \end{array} & \begin{array}{c} \ldots \\ \ldots \end{array} & \begin{array}{c} 0\\ 0 \end{array}\\ \vdots & \ddots & \vdots \\ \begin{array}{cc} 0 & 0 \end{array} & \cdots & X_{N} \end{array}\right]\left[\begin{array}{c} \beta_{1}\\ \beta_{2}\\ \vdots \\ \beta_{N} \end{array}\right]+\left[\begin{array}{c} u_{1}\\ u_{2}\\ \vdots \\ u_{N} \end{array}\right]$$

## 3. Results

### Performance of Subjects

The mean values of the performance factors of LPGA and KLPGA golfers by year are presented in Table 3 and Table 4.

Table 3 outlines the descriptive statistics for the LPGA golfers’ performance in the 3-year period from 2018 to 2020. Golfers in the top 60 of prize money rankings were selected for analysis, for a total of 178 golfers after excluding 2 cases who had missing values in their data. The specific performance factors are as follows: SA was lowest in 2019 at 70.73, DD was longest in 2019 at 259.68 yards, and DA was most accurate in 2018 at 73.59; GIR was highest in 2019 at 72.05, PA was highest in 2020 at 29.97, and SS was highest in 2018 at 49.14; and BIR was highest in 2018 at 309.25, EAG was highest in 2019 at 7.24, 60SA was highest in 2019 at 29.66, and T10 was highest in 2019 at 2.86.

Table 4 outlines the descriptive statistics for the KLPGA golfers’ performance in the 3-year period from 2018 to 2020. Golfers in the top 60 of prize money rankings were selected for analysis, amounting to a total of 180 golfers; there were no exclusions due to missing values. The specific performance factors are as follows: SA was lowest in 2018 at 71.70, DD was longest in 2018 at 242.42 yards, and DA was most accurate in 2020 at 75.87; GIR was highest in 2018 at 73.01, PA was highest in 2019 at 30.48, BIR was highest in 2018 at 17.46, and RE was highest in 2018 at 59.31; and PS was highest in 2018 at 85.55, PB was highest in 2018 at 17.59, and T10 was highest in 2019 at 22.23.

The following analysis was conducted to establish whether there were differences in the LPGA and KLPGA golfers’ prize money ranking factors in 2018. Table 5 shows that the regression model for the prize money ranking of LPGA and KLPGA golfers produced a statistically significant result (*p* < 0.001). For the LPGA golfers, the factors of PA (B = 18.179, *p* < 0.001) and SA (B = 4.363, *p* < 0.05) had statistically significant positive (+) effects on prize money, while GIR (B = −3.093, *p* < 0.001) and BIR (B = −0.119, *p* < 0.05) had statistically significant negative (−) effects on prize money. Regarding the KLPGA golfers, there were no statistically significant performance factors in their prize money rankings (*p* > 0.05). In addition, the test for the effect of the performance factors on prize money rankings indicated that PA showed a statistically significant difference (χ^2^ = 4.17, *p* < 0.05), with LPGA golfers having a larger influence.

The following analysis was conducted to establish whether there were differences in the LPGA and KLPGA golfers’ prize money ranking factors in 2019. Table 6 shows that the regression model for the LPGA and KLPGA golfers’ prize money rankings produced a statistically significant difference (*p* < 0.001). For the LPGA golfers, the performance factors indicated that SA (B = 18.112, *p* < 0.05) had a statistically significant positive (+) effect on prize money, while T10 (B = −1.680, *p* < 0.01) had a statistically significant negative (−) effect on prize money. Regarding the KLPGA golfers, DD (B = −0.234, *p* < 0.05) had a statistically significant negative (−) effect on their prize money rankings. In addition, the test for the effect of the performance factors on prize money rankings indicated that T10 showed a statistically significant difference (χ^2^ = 4.10, *p* < 0.05), with LPGA golfers experiencinga larger influence.

The following analysis was conducted to establish whether there were differences in the LPGA and KLPGA golfers’ prize money ranking factors in 2020. Table 7 shows that the regression model for the LPGA and KLPGA golfers’ prize money rankings produced a statistically significant difference (*p* < 0.001). For the LPGA golfers, no performance factors were statistically significant in their prize money ranking. For the KLPGA golfers, DD (B = −0.937, *p* < 0.001) and T10 (B = −0.266, *p* < 0.01) had statistically significant negative (−) effects on prize money. The test of the performance factors’ effect on prize money rankings indicated that DD showed a statistically significant difference (χ^2^ = 13.25, *p* < 0.001), with KLPGA golfers experiencinga larger influence.

## 4. Discussion

In all sporting events, athletes enhance their performance to achieve victory. Specifically, as golf is a sport that cannot be won with a mere handful of factors, golfers must systematically and scientifically hone their skills using training programs that are tailored to their performance outcomes. The current research used the most recent (2018–2020) performance outcome data on the top 60 golfers in the prize money rankings of both the LPGA and KLPGA to analyze the skill factors, skill outcomes, and outcome factors for eachseason.

First, for the LPGA golfers in 2018, the prize money ranking performance factors were PA, SA, GIR, and BIR. This is consistent with the finding in [29] that suggested PA and GIR as performance-enhancing factors and the result in [12] that indicated PA, GIR, and BIR as factors in athletic performance. The results of the current study suggest that PA has the largest impact on golfers’ performance, with a decrease of onein PA associated with an increase of approximately 18.179 in prize money ranking; similarly, a decrease of onein SA is associated with an increase of 4.363 in prize money ranking; an increase of 1% in GIR is associated with an increase of 3.093 in prize money ranking; and an increase by one in BIR is associated with an increase of 0.119 in prize money ranking, suggesting that SA is the most important factor. That is, for the 2018 LPGA, it has been verified that accurate putting has the strongest association with prize money ranking. In contrast, no performance factors were statistically significant in the golfers’ prize money rankings in the 2018 KLPGA, with PA showing a statistically significant difference in its effect on the performance of the two groups (LPGA and KLPGA golfers). In 2018, PA had a larger influence on the prize money ranking of LPGA golfers than on KLPGA golfers. For every decrease of 1 in PA, there was an increase of approximately 18.179 in the prize money ranking for the LPGA [11], thereby supporting the findings of the current research. This is hypothesized to be caused by the fact that the course length of the LPGA is approximately 141 yards longer than that of the KLPGA [13]. For PA, the influence on the LPGA golfers was larger than on the KLPGA golfers. These results indicate that PA is an important factor with a decisive influence on the outcomes of the 2018 LPGA.

Second, for the LPGA golfers in 2019, the prize money ranking performance factors were SA and T10; for the KLPGA golfers, DD affected their prize money rankings. This supports [30]’s findings, which indicated that the number of rounds in the 60s, along with T10, led to sustained performance outcomes for high-ranking golfers in the 2016 LPGA prize money rankings. Regarding the KLPGA, [13] indicated that DD and T10 affected the prize money rankings of the 2019 KLPGA, which is consistent with the results of the current study. There was a difference in the impact of T10 on the 2019 prize money rankings between the two groups of golfers, with a larger influence on the LPGA golfers than on the KLPGA golfers. At the KLPGA in 2019, the winner was different for every event, and there were significant changes in the top-10 finish list; however, in the LPGA, there was less fluctuation in the Top-10 list, since there were 38 golfers who made it to the list in two or more events and 13 golfers who made it to the list in five or more events. Because of such differences, it seems that T10 had a larger influence on the prize money rankings of the LPGA than the KLPGA.

Third, no performance factors were statistically significant in the LPGA golfers’ prize money rankings in 2020; for the KLPGA, DD and T10 had statistically significant impacts on their prize money rankings. These were also indicated to be important factors for KLPGA golfers in 2019, suggesting that DD is a crucial factor for KLPGA golfers. It was suggested by [29] that DD differentiated the top 10 golfers’ performance outcomes from those of middle- and low-ranking golfers, supporting the findings of the current study. An evaluation of the differences in the performance factors between LPGA and KLPGA golfers in 2020 indicated statistically significant differences in DD, which had a larger influence on the KLPGA golfers than on the LPGA golfers. Regarding such results, it is hypothesized that DD started exerting important influence on the prize money rankings of the 2020 KLPGA due to the following factors: compared to 2019, the course length of the 2020 KLPGA decreased from 5570 yards to 5558 yards, while the level of difficulty increased in 2020 compared to 2019. This result suggests that KLPGA golfers requiretraining programs to enhance their DD.

Finally, a comparative analysis of the differences in performance factors between the LPGA and KLPGA golfers from 2018 to 2020 indicated that the LPGA golfers’ PA was higher in 2018, the LPGA golfers’ T10 was higher in 2019, and the KLPGA golfers’ DD was higher in 2020. This confirms that crucial performance factors for golfers can vary by season and by region. That is, the short game is more important for LPGA Tours, while the shot distance tends to be more important for KLPGA Tours.

## 5. Conclusions and Recommendations: Skill Factors, Skill Outcomes, Outcome Factors by Season

To summarize the results, between 2018 and 2020, it became difficult to determine the LPGA golfers’ prize money ranking performance factors, while the shot distance became a more important factor for the KLPGA golfers compared to their short game skills. Prior to the scaling down of the tour tournaments due to COVID-19, LPGA golfers focused on their short game skills; as the tours were reduced to half their original magnitude after COVID-19 and international movement became restricted, it became difficult to discern the performance factors in LPGA golfers’ prize money rankings. However, as the COVID-19 pandemic is entering a stable phase and both the LPGA and KLPGA are recovering their prepandemic scales, it is necessary to verify the performance factors that are required for KLPGA golfers who aspire to compete on the global stage. In the current research, the caliber required of LPGA golfers has been identified and the suggestion made that golfers should increase their short game abilities. To achieve this, it will be beneficial for KLPGA golfers to decrease their average number of putts or increase the accuracy of their iron shots. Instead, as golfers who were active on the LPGA stage returned to the KLPGA, they ended up competing in relatively shorter courses [27], which had a crucial impact on the golfers’ prize money rankings in the KLPGA.

As the COVID-19 crisis persists, it seems that golfers’ prize money ranking performance factors will change in the future. The current study is significant in that it is the first of its kind to compare the conditions before and after the COVID-19 crisis. The practical implications of the current study include the fact that additional analyses of LPGA Tour golfers are required to establish the factors that are critical for performance outcomes, and that the KLPGA golfers requiretraining programs to increase their shot distance.

Finally, the statistical data from these tours can serve important roles in evaluating the strengths and weakness of LPGA and KLPGA golfers and in analyzing each individual golfer’s performance deficiencies. In addition, the tours provide objective data that help to isolate individual golfers’ performance areas that require intensive training. Follow-up studies should focus on finding female golfers’ most influential performance factors in the new normal age through comparative pre-post analyses of KLPGA and LPGA golfers’ performance outcomes.

The current research aggregated data from 2018 to 2020 for analysis; however, due to the COVID-19 crisis, the amountof data for 2020 was not consistent due to the significant decrease in the number of competitions. Future studies that combine the pre-andpost-COVID-19 data for analysis will be able to present a more accurate picture of the performance factors required for domestic female golfers aspiring to compete on the international stage.

## Figures and Tables

**Table 1 sports-10-00072-t001:** Definition of key performance indicators(LPGA).

Independent Variables	Calculus
Scoring Average (SA)	(Total number of strokes * 72/Par)/rounds
Driver Distance (DD)	Average driver distance
Driver Accuracy (DA)	The accuracy of the driver’s ball
Greens in Regulation (GIR)	Total number of greens in regulation/holes * 100
Putting Average (PA)	Total number of putting/rounds
Sand Save (SS)	refers to getting up and down out of a green side bunker
Birdies (BIR)	Total number of birdies
Eagles (EAG)	Total number of eagles
60-strokes Average (60SA)	Total number of rounds in the 60′s
Top-10 Finish (T10)	Total number of top-10 finish/tournaments * 100

**Table 2 sports-10-00072-t002:** Definition of key performance indicators(KLPGA).

Independent Variables	Calculus
Scoring Average (SA)	(Total number of stroke * 72/Par)/rounds
Driver distance (DD)	Average driver distance
Driver Accuracy (DA)	The accuracy of the driver’s ball
Greens in Regulation (GIR)	Total number of greens in regulation/holes * 100
Putting Average (PA)	Total number of putting/rounds
Birdies (BIR)	Total number of birdies
Recovery (RE)	Percentage of scores better than par (par or birdie)
Par Saves (PS)	The percentage of scores better than par
Par Breaks (PB)	Buddy and the percentage of people who score better than that
Top-10 Finish (T10)	Total number of top-10 finish/tournaments * 100

**Table 3 sports-10-00072-t003:** Performance of LPGA players (2018–2020) (M ± SD).

Independent Variables	2018	2019	2020
Scoring Average (SA)	70.95 ± 0.79 (69.4~74.4)	70.73 ± 0.68 (69.1~72.2)	71.25 ± 0.72 (68.7~72.8)
Driver Distance (DD)	253.72 ± 17.00 (146.5~289.5)	259.68 ± 8.55 (244.7~277.5)	255.66 ± 9.46 (237.4~286.6)
Driver Accuracy (DA)	73.59 ± 5.80 (52.2~85.7)	73.36 ± 5.38 (62.0~84.1)	74.95 ± 4.99 (66.6~93.4)
Greens in Regulation (GIR)	70.78 ± 3.04 (63.9~77.0)	72.05 ± 2.94 (66.6~79.6)	68.86 ± 2.95 (61.4~77.6)
Putting Average (PA)	29.61 ± 0.47 (28.4~30.6)	29.82 ± 0.50 (27.6~30.8)	29.97 ± 0.50 (28.6~30.8)
Sand Save (SS)	49.14 ± 7.17 (32.9~63.7)	47.94 ± 7.37 (31.4~62.5)	48.53 ± 8.73 (31.6~71.4)
Birdies (BIR)	309.25 ± 51.63 (182.0~470.0)	306.71 ± 51.98 (174.0~418.0)	156.32 ± 35.42 (52.0~208.0)
Eagles (EAG)	6.27 ± 5.42 (0.0~40.0)	7.24 ± 4.05 (0.0~23.0)	2.43 ± 1.60 (0.0~6.0)
60-strokes Average (60SA)	28.75 ± 8.62 (15.0~57.0)	29.66 ± 8.90 (13.0~52.0)	13.52 ± 4.37 (3.0~23.0)
Top-10 Finish (T10)	2.63 ± 2.33 (0.0~8.0)	2.86 ± 2.29 (0.0~8.0)	2.77 ± 2.14 (0.0~8.0)

**Table 4 sports-10-00072-t004:** Performance of KLPGA players (2018–2020) (M ± SD).

Independent Variables	2018	2019	2020
Scoring Average (SA)	71.70 ± 0.81 (69.9~73.0)	71.97 ± 1.00 (69.0~74.5)	71.79 ± 0.87 (69.6~73.0)
Driver Distance (DD)	242.42 ± 6.93 (230.5~259.2)	240.11 ± 8.67 (219.8~262.5)	238.48 ± 7.58 (222.0~259.5)
Driver Accuracy (DA)	75.83 ± 4.38 (65.8~85.3)	75.26 ± 4.62 (61.5~84.3)	74.71 ± 6.41 (47.4~85.2)
Greens in Regulation (GIR)	73.01 ± 3.79 (63.6~81.2)	71.74 ± 4.35 (61.5~82.7)	74.02 ± 3.09 (68.1~84.1)
Putting Average (PA)	30.44 ± 0.66 (27.8~31.8)	30.48 ± 0.65 (27.8~31.6)	30.71 ± 0.51 (29.2~31.9)
Birdies (BIR)	17.46 ± 2.42 (12.9~25.3)	16.86 ± 3.05 (12.3~30.6)	17.02 ± 2.27 (12.5~21.9)
Recovery (RE)	59.31 ± 4.69 (48.0~68.3)	58.65 ± 5.65 (45.2~80.4)	57.92 ± 4.98 (49.0~69.1)
Par Saves (PS)	85.55 ± 2.15 (81.6~90.2)	84.87 ± 2.76 (77.5~94.0)	85.45 ± 2.58 (80.6~91.8)
Par Breaks (PB)	17.59 ± 2.45 (13.0~25.3)	16.99 ± 3.05 (12.4~30.6)	17.16 ± 2.28 (12.8~22.0)
Top-10 Finish (T10)	21.43 ± 15.76 (3.7~75.0)	22.23 ± 18.95 (0.0~100.0)	20.35 ± 18.87 (0.0~87.5)

**Table 5 sports-10-00072-t005:** Performance effect of LPGA vs. KLPGA on Rank (2018).

Independent Variables	LPGA			KLPGA			χ^2^
	Coef.	S.E.	Z	Coef.	S.E.	Z	
Scoring Average (SA)	4.363	1.821	2.400 *	7.512	13.530	0.560	0.05
Driver Distance (DD)	0.040	0.087	0.470	−0.496	0.398	−1.250	1.73
Driver Accuracy (DA)	0.166	0.230	0.720	−0.813	0.586	−1.390	2.41
Greens in Regulation % (GIR)	−3.093	0.664	−4.660 ***	3.344	3.276	1.020	3.71
Putting Average (PA)	18.179	3.466	5.250 ***	−13.613	15.174	−0.900	4.17 *
Birdies (BIR)	−0.119	0.055	−2.170 *	4.565	10.119	0.450	0.21
Top-10 Finish % (T10)	−1.059	0.798	−1.330	−0.184	0.195	−0.940	1.14
F (*p*)	9.98 (<0.001)			11.90 (<0.001)			
R^2^	0.675			0.708			
Adjusted R^2^	0.608			0.649			

* *p* < 0.05, *** *p* < 0.001.

**Table 6 sports-10-00072-t006:** Performance effect of LPGA vs. KLPGA on rank (2019).

Independent Variables	LPGA			KLPGA			χ^2^
	Coef.	S.E.	Z	Coef.	S.E.	Z	
Scoring Average (SA)	18.112	7.346	2.470 *	21.451	15.273	1.400	0.04
Driver Distance (DD)	−0.267	0.273	−0.980	−0.234	0.117	−2.000 *	0.01
Driver Accuracy (DA)	0.030	0.388	0.080	0.607	0.423	1.430	1.01
Greens in Regulation % (GIR)	0.634	1.419	0.450	0.486	3.955	0.120	0.00
Putting Average (PA)	−4.436	6.720	−0.660	−0.238	17.646	−0.010	0.05
Birdies (BIR)	−0.055	0.050	−1.100	−9.175	11.431	−0.800	0.64
Top-10 Finish% (T10)	−1.680	0.633	−2.660 **	−0.340	0.195	−1.750	4.10 *
F (*p*)	9.92 (<0.001)			10.60 (<0.001)			
R^2^	0.674			0.688			
Adjusted R^2^	0.606			0.623			

* *p* < 0.05, ** *p* < 0.01.

**Table 7 sports-10-00072-t007:** Performance effect of LPGA vs. KLPGA on Rank (2020).

Independent Variables	LPGA			KLPGA			χ^2^
	Coef.	S.E.	Z	Coef.	S.E.	Z	
Scoring Average (SA)	12.830	13.033	0.980	20.591	15.364	1.340	0.15
Driver Distance (DD)	0.026	0.186	0.140	−0.937	0.189	−4.970 ***	13.25 ***
Driver Accuracy (DA)	0.349	0.565	0.620	0.014	0.185	0.070	0.32
Greens in Regulation % (GIR)	−0.389	2.931	−0.130	−1.429	1.973	−0.720	0.09
Putting Average (PA)	3.133	12.867	0.240	6.481	9.191	0.710	0.04
Birdies (BIR)	−0.133	0.095	−1.410	−6.431	6.377	−1.010	0.97
Top-10 Finish % (T10)	−0.908	0.794	−1.140	−0.266	0.094	−2.830 **	0.65
F (*p*)	4.89 (<0.001)			19.98 (<0.001)			
R^2^	0.500			0.803			
Adjusted R^2^	0.397			0.763			

** *p* < 0.01, *** *p* < 0.001.

## Data Availability

The public data was found here: 2020 Summer Olympics-Wikipedia https://en.wikipedia.org/wiki/2020_Summer_Olympics, accessed on 9 December 2021).

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
