# Peer review of "Comparative Analysis of Performance Factors between Ladies Professional Golf Association and Korea Ladies Professional Golf Association Golfers"

_sports, 2022, doi:10.3390/sports10050072_

Round 1

Reviewer 1 Report

The main purpose of the work was to analyze the performance factors of both the Ladies Professional Golf Association and the Korea Ladies Professional Golf Association players, and suggest which performance factors they should improve to play in the world level games. 

The paper is generally well written based on sound literature, the results well presented and discussed with respect to the literature.

The work is written following the steps of the scientific method.

The conclusions are the answer to the research question and result from the conducted research.

The study is well designed. However I have some minor comments I’d like to express.

The Abstract must be improved, with a sequence of the following systematization: Objectives, Methods, Results, and Conclusions.

While the introduction is written extensively and properly introduces the research topic, I do suggest extending the information on performance factors in the sport being described. In this way, the purpose of the research will be even more emphasized.

At the beginning of the introduction, try to define the inclusion and exclusion criteria for the analysis.

Provide a brief summary of the limitations of the evidence included in the review.

Perhaps the abbreviations should already be explained in the abstract. 

Explain the remaining abbreviations in the text, eg SPGA. 

The discussion can be slightly expanded.

It is worth adding a practical (appilcative) conclusion.

In my opinion, the conclusions should be more specific. not generalized, but this is only a suggestion.

The article is generally valuable and correctly written, please treat the above comments only as suggestions.

Author Response

Reviewer 1

1. The Abstract must be improved, with a sequence of the following systematization: Objectives, Methods, Results, and Conclusions. >p.1 Improvements have been made to the Objectives-Methods-Results-Conclusions, following the sequence.

2. While the introduction is written extensively and properly introduces the research topic, I do suggest extending the information on performance factors in the sport being described. In this way, the purpose of the research will be even more emphasized. >p.3 The final portion of the Introduction section has been supplemented.

3. At the beginning of the introduction, try to define the inclusion and exclusion criteria for the analysis. >I am having trouble understanding this comment. I would be grateful if you could give some example studies.

4. Provide a brief summary of the limitations of the evidence included in the review. >p.9 Supplementations have been made to the limitations section of the study.

5. Perhaps the abbreviations should already be explained in the abstract. >p.1 The abbreviations have been explained in the abstract.

6. Explain the remaining abbreviations in the text, eg SPGA. >p.1, 3 The explanation for terms such as COVID-19 and SPGA has been added with the initial usage of the terms.

7. The discussion can be slightly expanded. >p.7-8 It has been expanded.

8. It is worth adding a practical (appilcative) conclusion. >p.9 Specific training methods for KLPGA golfers have been suggested: increasing putting abilities, and training for iron shots.

Thank you.

Reviewer 2 Report

Thanks to the author for the valuable manuscript. Statistical analysis of large volumes of data provides a better understanding of the sports process.

The abbreviation LPGA and KLPGA should not be used in the title and keywords of the article. It is necessary to explain the special terms before submitting them.

The introduction does not adequately reflect the importance of the problem. I would like the author to explain level the significance of the results and how results can be used.

In the introduction, the author does not provide a clear answer as to why it is important to identify and evaluate the factors directly related to professional golfers ’skills, analyze those with an influence on prize money, and predict those that apply to individual golfers.

Part of the introduction analyzes has information that is not directly related to the problem being addressed that can be omitted.

When evaluating the description of the methodology, I lacked a detailed data processing algorithm. It is not enough to indicate two sources, it is necessary to provide data processing tools (formulas) and criteria for their application.

The author does not provide examples of primary data. I think it is necessary to do that.

In the section Conclusion and Recommendations, the author discusses things that are not the object of the research: skill factors, skill outcomes and outcome factors by season. I think this section requires a major overhaul, linking it to the purpose of the research.

Author Response

Reviewer 2

  1. The abbreviation LPGA and KLPGA should not be used in the title and keywords of the article. It is necessary to explain the special terms before submitting them. >p.1 I have written the full name for the title and keywords.

2. The introduction does not adequately reflect the importance of the problem. I would like the author to explain level the significance of the results and how results can be used. In the introduction, the author does not provide a clear answer as to why it is important to identify and evaluate the factors directly related to professional golfers ’skills, analyze those with an influence on prize money, and predict those that apply to individual golfers. >p.3 Supplementations have been made to the final paragraph of the Introduction section.

3. Part of the introduction analyzes has information that is not directly related to the problem being addressed that can be omitted. >I assume that you are addressing the sentences that were written for better flow (e.g. COVID-19, vitalization of the golf industry, factors unrelated to a performance that affect the prize money ranking) of the paragraph. I have added these for better flow. If this is not what you had meant, please address it with follow-up comments.

4. When evaluating the description of the methodology, I lacked a detailed data processing algorithm. It is not enough to indicate two sources, it is necessary to provide data processing tools (formulas) and criteria for their application. >p.4 I have added descriptions regarding the data collection method and statistical analysis software.

5. The author does not provide examples of primary data. I think it is necessary to do that. >The author has aggregated data on individual golfers—provided by the LPGA and KLPGA—using Excel, and used it as the primary data. Please refer to the attached files.

6. In the section Conclusion and Recommendations, the author discusses things that are not the object of the research: skill factors, skill outcomes and outcome factors by season. I think this section requires a major overhaul, linking it to the purpose of the research. >p.8-9 Supplementations have been made regarding the yearly differences.

Thank you.

Reviewer 3 Report

Dear author,

I have carefully analysed the manuscript and I suggest you to consider the following recommendations:

  1. "Such data are used to closely analyze teams’ and individual team members’ athletic performances." requires one or more references.
  2. "the records of professional golfers' performance in the PGA, LPGA, and SPGA by [18]" SPGA acronym must be made explicit.
  3. "research factors" within the text and the tables should be replaced by "key performance indicators" or "KPIs" that are the most frequently used terms in performance analysis field.
  4. What is the criterion for selecting the KPIs? Previous references, experts' opinion? Please clarify this point.
  5. How the data was stored/managed (eg. databases, excel files) and what software did you use to implement the analysis? Please add this information.
  6.  Please add a statistical dispersion index, such as standard deviation, range, ecc., for the descriptive data in table 3 and 4.

Author Response

Reviewer 3

1. "Such data are used to closely analyze teams’ and individual team members’ athletic performances." requires one or more references. >p.2 Three existing research articles have been referenced.

2.  "the records of professional golfers' performance in the PGA, LPGA, and SPGA by [18]" SPGA acronym must be made explicit. >Supplementations have been made on page1 and 3.

3. "research factors" within the text and the tables should be replaced by "key performance indicators" or "KPIs" that are the most frequently used terms in performance analysis field. >p.4 "Research factors" in Tables 1 and 2 have been replaced with "key performance indicators".

4. What is the criterion for selecting the KPIs? Previous references, experts' opinion? Please clarify this point. >p.4 I have referenced existing research that used the same KPI that was used in the current study.

5. How the data was stored/managed (eg. databases, excel files) and what software did you use to implement the analysis? Please add this information. >p.4 The source of data, method of organization, and name of analysis software have been added to the Analysis section of the Methods.

6. Please add a statistical dispersion index, such as standard deviation, range, ecc., for the descriptive data in table 3 and 4. >p.5-6 The standard deviation and range have been added.

Thank you.
